# Evaluation of Subtle Auditory Impairments with Multiple Audiological Assessments in Normal Hearing Workers Exposed to Occupational Noise

**DOI:** 10.3390/brainsci13060968

**Published:** 2023-06-19

**Authors:** Alessandra Giannella Samelli, Clayton Henrique Rocha, Mariana Keiko Kamita, Maria Elisa Pereira Lopes, Camila Quintiliano Andrade, Carla Gentile Matas

**Affiliations:** Department of Physical Therapy, Speech-Language-Hearing Sciences, and Occupational Therapy, Medical School (FMUSP), University of São Paulo, São Paulo 05360-160, SP, Brazil; claytonhr@usp.br (C.H.R.); marianakeiko@usp.br (M.K.K.); fgamelisa@gmail.com (M.E.P.L.); miquintiliano1@gmail.com (C.Q.A.); cgmatas@usp.br (C.G.M.)

**Keywords:** noise, occupational, cochlea, auditory perceptual disorders, hearing tests

## Abstract

Recent studies involving guinea pigs have shown that noise can damage the synapses between the inner hair cells and spiral ganglion neurons, even with normal hearing thresholds—which makes it important to investigate this kind of impairment in humans. The aim was to investigate, with multiple audiological assessments, the auditory function of normal hearing workers exposed to occupational noise. Altogether, 60 workers were assessed (30 in the noise-exposure group [NEG], who were exposed to occupational noise, and 30 in the control group [CG], who were not exposed to occupational noise); the workers were matched according to age. The following procedures were used: complete audiological assessment; speech recognition threshold in noise (SRTN); speech in noise (SN) in an acoustic field; gaps-in-noise (GIN); transient evoked otoacoustic emissions (TEOAE) and inhibitory effect of the efferent auditory pathway; auditory brainstem response (ABR); and long-latency auditory evoked potentials (LLAEP). No significant difference was found between the groups in SRTN. In SN, the NEG performed worse than the CG in signal-to-noise ratio (SNR) 0 (*p*-value 0.023). In GIN, the NEG had a significantly lower percentage of correct answers (*p*-value 0.042). In TEOAE, the NEG had smaller amplitude values bilaterally (RE *p*-value 0.048; LE *p*-value 0.045) and a smaller inhibitory effect of the efferent pathway (*p*-value 0.009). In ABR, the NEG had greater latencies of wave V (*p*-value 0.017) and interpeak intervals III-V and I-V in the LE (respective *p*-values: 0.005 and 0.04). In LLAEP, the NEG had a smaller P3 amplitude bilaterally (RE *p*-value 0.001; LE *p*-value 0.002). The NEG performed worse than the CG in most of the assessments, suggesting that the auditory function in individuals exposed to occupational noise is impaired, even with normal audiometric thresholds.

## 1. Introduction

Noise-induced hearing loss results from lesions or dysfunctions, especially in the cochlear outer hair cells; these cells were believed to be the structures most vulnerable to noise. Thus, noise exposure that did not cause permanent hearing loss used to be considered safe [1,2,3].

However, recent studies have demonstrated that the structures most vulnerable to noise following exposure that leads to temporary hearing loss are the synapses between the inner hair cells and the spiral ganglion neurons, even with normal hearing thresholds. This synaptopathy has been demonstrated in guinea pigs and is named hidden hearing loss [2,3,4,5].

Furthermore, this synaptopathy is specifically located in the fibers that form synapses on the modiolar side of the inner hair cells, whose spontaneous rates are relatively low. They are particularly important in noisy environments because they are more resistant to masking by continuous background noise due to their higher thresholds. Therefore, high spontaneous rate fibers (the pillar side of inner hair cells) are believed to be the ones that contribute the most in quiet environments, but as the background noise increases, low spontaneous rate fibers are particularly important. Hence, in situations of greater auditory demand, with higher background noise, speech intelligibility and temporal auditory processing may be impaired, and it has been suggested that this is a sign of hidden hearing loss [3,4,5,6,7].

Reduced input, with or without elevated hearing thresholds, may also lead to central plasticity because the input to the auditory nerve decreases, causing a cascade of events that impair the whole auditory pathway [3]. As this cascade of events can cause changes throughout the entire auditory system, including the peripheral and central auditory pathways, special assessments, such as long-latency evoked potentials (LLAEP), should be used to verify the most rostral portion of the auditory system.

It is important to highlight that there is no consensus regarding the occurrence of noise-induced synaptopathy in humans. Despite advances in the comparisons between animal and human models, further studies in humans are still needed to establish strong and reliable conclusions, especially regarding the diagnostic tools [8].

The studies on noise-induced hidden hearing loss suggest that humans are less vulnerable to noise damage than smaller mammals. Moreover, animals and humans are exposed to different types of noise. Guinea pigs are usually exposed to continuous noise at the highest possible level; this noise does not cause PTS, but it does cause synaptic losses in a short period of time. Humans, however, do not experience this in real life; instead, they are often exposed to intermittent noise at a much lower level and for a long time, especially when hearing protection devices are used according to the current security standards [9,10,11].

Considerable attention has been given to experiments on specific mice strains to identify candidate assays using non-invasive measures that could potentially identify hidden hearing loss in humans. Nonetheless, there is still no consensus on which part of the neurophysiological evidence observed in animal models would also be evident in humans; this underlies the cause of the difficulties in speech intelligibility experienced by subjects with normal audiograms after occupational noise exposure [12].

Therefore, it is highly important to investigate the feasibility and inclusion of multiple assessments (besides conventional audiometry) to monitor workers exposed to noise. Hearing thresholds are normal in cases of hidden hearing loss, and these workers may present subtle auditory impairments, such as difficulty in following a conversation in noisy environments. Moreover, this type of impairment must be confirmed in humans, due to its possible practical consequences, including occupational regulations regarding noise exposure limits.

Our study hypothesizes that workers exposed to noise, even while wearing hearing protection, may have an unnoticed auditory impairment, which can be detected with multiple peripheral and central auditory assessments. The noise-exposure and control groups were matched according to age and underwent multiple audiological procedures to assess different portions of the peripheral and central auditory pathways.

## 2. Materials and Methods

This observational cross-sectional study was approved by the institution’s Ethics Committee under number 2.435.259/2017. The participants were informed in advance about the study, and those who agreed with it signed an informed consent form.

### 2.1. Sample

Altogether, 60 male individuals participated in the study; they were divided into two groups and were matched according to age (≤2 years): the noise-exposure group (NEG), comprising 30 individuals exposed to occupational noise, with a mean age of 35.6 ± 7.09 years (min: 23; max: 50), and the control group (CG), comprising 30 individuals not exposed to occupational noise, with a mean age of 35.37 ± 7.56 years (min: 22; max: 49).

The inclusion criteria for the study participants were as follows: age between 18 and 50 years; normal hearing thresholds (<25 dBHL at 250 to 8000 Hz); absence of changes in the external auditory meatus; no exposure to chemical products; absence of tinnitus; not undergoing chemotherapy or radiotherapy; not taking any ototoxic drugs; exposure to occupational noise for at least 5 years (NEG); no exposure to occupational noise (CG); and no exposure to non-occupational noise (both groups). Both groups’ conventional and high-frequency audiometry results can be found in Appendix A.

The workers were selected based on the university’s Environmental Risks Prevention Program, which describes the risks to which each worker is exposed in their workday. The NEG comprised individuals who worked in maintenance at the university, and the CG comprised individuals who worked in other university departments without noise (basically, administrative staff). The NEG participants had been exposed to intermittent noise (Lavg [average sound pressure level over a period of time]: 88 dBA; minimum: 75 dBA, maximum: 111 dBA; 69% of the daily dose) during their 8 h workday for a mean of 8.6 years (SD: 6.1 years) in which they had been in that position (exposure). All of them used hearing protection during their workday.

### 2.2. Procedures

The tests described below were reproduced in a sound booth with a two-channel audiometer (model AC-40; Interacoustics, Middelfart, Denmark).

−Sentence recognition threshold in noise (SRTN) was assessed with a list of sentences in Portuguese (LSP) [13]—a list of sentences was presented to individuals in ipsilateral white noise through TDH-50 earphones. They were asked to repeat the sentences as they understood them. The noise was maintained at 65 dBHL throughout the test, while the sentences were initially presented at 68 dBHL. After each sentence that the patient repeated correctly, the intensity was decreased by 4 dB; then, after the first mistake, 2 dB intervals were used for either correct or wrong answers. Thus, the SRTN was obtained which corresponded to the final signal-to-noise ratio (SNR).−Speech recognition in noise (SN) was assessed with monosyllables, using an acoustic field system with three loudspeakers. Pink noise was presented in channel 1, which was connected to the loudspeakers on the participants’ right and left sides, 85 cm away from them and at a 90° azimuth. Speech stimuli were presented in channel 2, which was connected to the central loudspeaker, placed 100 cm away from them and at a 0° azimuth. The study used recorded monosyllables [14,15] as stimuli, presenting 25 monosyllables in each SNR (0 and −10 dB), while the speech remained at 65 dB(A). The results were based on the percentage of correct answers in each SNR.−Temporal resolution was assessed with the gaps-in-noise test (GIN)—the GIN test [16] was used and presented stimuli monaurally through TDH-50 earphones 50 dBSL above the mean pure-tone thresholds of the right ear at 500, 1000, and 2000 Hz. The results were based on the number of silent intervals correctly detected, with the threshold set as the shortest silent interval perceived by the individual in 4/6 presentations. The normal criteria used in this study were 70% or more correct answers and a 5 ms or lower gap-detection threshold [17].−An analysis of the transient evoked otoacoustic emissions (TEOAE) and the inhibitory effect of the efferent auditory pathway was conducted. Nonlinear stimuli were presented at 80 dBSPL to assess responses at 1 kHz, 1.4 kHz, 2 kHz, 2.8 kHz, and 4 kHz and at the total response amplitude (ILO 292; Otodynamics, Hatfield, UK). They were considered present when the SNR was ≥3 dBSPL at three consecutive frequencies [18]. The participants whose responses were present in the TEOAE had the inhibitory effect of the efferent auditory pathway assessed with linear click stimuli at 60 dBSPL in white noise (also at 60 dBSPL), presented with 10 s intervals between them. To calculate the inhibitory effect, the TEOAE results with and without noise were transformed into micropascals (µPa); then, the difference between the emissions in these two situations was obtained. Differences with positive values were considered to represent the presence of the inhibitory effect of the efferent auditory pathway [19]. The percentage of this inhibitory effect was also calculated. As the right ear has an advantage over the left one with regard to the inhibitory effect of the efferent pathway [20], only the right ear results were considered for analysis.−Amplitudes and latencies were assessed with auditory brainstem response (ABR)—electrodes were positioned on the vertex (Cz), forehead (FPz), and left (M1) and right mastoid (M2), with impedance values below 5 kiloohms (kΩ). The integrity of the auditory pathways was assessed, first in the right ear, then in the left one, with click stimuli presented through insert headphones at 80 dBnHL (Smart EP, Intelligent Hearing System). This protocol used a presentation rate of 19 clicks per second, each one lasting 0.1 ms, totaling 2000 stimuli [21]. Two collections were made from each ear to confirm wave reproducibility. The amplitudes and latencies of waves I, III, and V and the interpeak intervals I-III, III-V, and I-V were analyzed according to the normal values set for the equipment [22]. The V/I amplitude ratio was also analyzed.−Amplitudes and latencies were assessed with LLAEP; initially, the skin was cleaned with abrasive paste at the places where the electrodes were positioned: vertex (Cz), on the right (M2) and left (M1) mastoids, and on the forehead (Fpz). The electrode impedance values should be below 5 kOhms. Tone-burst stimuli were presented monaurally at 75 dBnHL, at a rate of 1.1 stimuli per second, totaling 300 stimuli; 15% were rare stimuli presented at 2000 Hz, while 85% were frequent stimuli presented at 1000 Hz (Smart EP, Intelligent Hearing System). The analysis window lasted 512 ms, and the high-pass and low-pass filters were from 1 to 30 Hz. The subjects were instructed to pay attention to the rare stimuli and mentally count every time they perceived them. In the tracing that resulted from subtracting the rare stimuli from the frequent ones (waveform subtraction), the P3 component was identified and analyzed regarding its latency and amplitude values (N2-P3). As for the tracing that corresponded to the frequent stimuli, the P1, N1, P2, and N2 components were identified and analyzed regarding their latency and amplitude values (P1-N1 and P2-N2) [22].

### 2.3. Data Analysis

The data were analyzed using descriptive measures (means, standard deviations, and minimum and maximum values) and inferential statistical tests (independent sample analysis of variance and chi-square test, when appropriate). The odds ratio (OR) and its confidence interval (CI) were also calculated for some tests. The significance level was set at 5% (*p* ≤ 0.05).

## 3. Results

The speech recognition index in silence did not differ between the groups (right ear: NEG = 99.5%; SD = 1.38; CG = 98.5%; SD = 2.67 − *p*-value = 0.073/left ear: NEG = 99.2%; SD = 1.63; CG = 99.3%; SD = 1.52 − *p*-value = 0.806). No statistically significant difference was found between the groups in the SRTN assessment. In the SN test, the NEG had a lower percentage of recognition in both SNRs, with a significant difference in SNR 0 (*p*-value 0.023) (Table 1).

In the GIN test, the gap detection threshold was higher in the NEG than the CG (5.86 ms and 5.20 ms, respectively), though with no significant difference. The percentage of correct answers was significantly lower in the NEG (64.1%). In general, the NEG performed worse as it had more individuals with abnormal GIN test results (43.3%) (Table 2). The OR of having an abnormal GIN test result was 3.059 (CI: 0.96–9.65) in the comparison between the individuals exposed to occupational noise and those not exposed.

Regarding TEOAE (Table 3), the NEG had a lower total amplitude than the CG, with statistically significant differences in both ears (*p*-value 0.048 in the RE; *p*-value 0.045 in the LE). Likewise, in the qualitative assessment the NEG had more individuals without responses than the CG (30% in the RE and 23.33% in the LE), with a significant difference in the RE (*p*-value 0.019).

In the assessment of the inhibitory effect of the efferent pathway in the RE, the NEG had a greater absence of the effect and less inhibition, with a significant difference in both situations (respective *p*-values: 0.015 and 0.009) (Table 4). Furthermore, the OR for the absence of the inhibitory effect of the efferent pathway was 8.1 (CI: 1.54–42.47) in the comparison between the individuals exposed to noise and those not exposed.

In ABR, the NEG had increased latency of wave V (*p*-value 0.017) and interpeak intervals III-V and I-V (respective *p*-values: 0.005 and 0.040) in the LE (Table 5).

Regarding LLAEP, no significant differences were found between the latencies analyzed. On the other hand, the NEG had a lower P3 amplitude in both ears (RE *p*-value 0.001 and LE *p*-value 0.002) (Table 6).

## 4. Discussion

The noise-exposed individuals in this study performed worse than the non-exposed ones in most of the assessments. The NEG performed worse than the CG in the speech-in-noise and GIN tests and had smaller TEOAE amplitudes, smaller inhibitory effects of the efferent system, increased ABR latencies in wave V and interpeak intervals I-V and III-V, and decreased P3 amplitudes.

Previous studies have suggested that the supraliminal effects of noise-induced hidden hearing loss can be perceived through auditory perception [2,4,6,23,24]. These effects may result from damage to the fibers with low trigger rates, thus explaining why audiologically normal individuals behave so differently when trying to listen to speech in noise [23].

This study assessed speech intelligibility in noise with SRTN (using earphones) and SN tests (presenting monosyllables in an acoustic field). However, the two groups performed differently only in SN, in SNR 0. Our hypothesis for these findings, with regard to the fact there was a difference between the groups in only one test, is that the SN test poses a more adverse condition for the auditory system; this makes clearer the NEG’s greater difficulty in listening to speech in noise. This difference in performance between normal hearing individuals exposed to occupational noise and those not exposed agrees with some of the previous studies in the literature, which, though using different methodologies, also verified worse assessment results in noise-exposed individuals [25,26,27].

Kumar et al. [25] assessed train engineers with regard to speech recognition in noise with a −5 dB SNR and verified a significantly worse performance in the noise-exposed group. The same was found by Hope et al. [26], who assessed speech perception in noise in pilots with a history of noise exposure. Liberman et al. [27] assessed normal hearing young people at high risk of synaptopathy and verified that this group performed worse in speech recognition in noise and time-compressed speech tests; they also had a longer reverberation time than the group at low risk of synaptopathy.

Another study, using a free-field system to emit monosyllables (65, 70, and 75 dB) and pink noise at different SNRs (0, −5, −10, and −15), compared a group exposed to occupational noise with a non-exposed group, both with normal hearing. They verified that the non-exposed group performed better in all situations, suggesting that this difference between the groups could be due to deficits in the auditory pathway caused by their history of noise exposure [28].

Nonetheless, other previous studies did not find any association between the history of noise exposure and worse performances in speech in noise and/or temporal processing tests [23,29]. This may have been influenced by the methodologies used in each study, such as the inclusion criteria, assessment methods, analysis criteria, speech and noise presentation levels, and so forth, which can impact the results. Furthermore, as pointed out by Pathasarathy et al. [24], variabilities in the studies using tests of speech recognition in noise may be related to bottom-up temporal processing abilities and top-down active listening resources; hence, it is difficult to generalize based on a single test type [24].

In temporal resolution assessment with the GIN test, the NEG had a higher gap detection threshold than the CG (5.86 ms and 5.20 ms, respectively). Consequently, they had a significantly lower percentage of correct answers (64.1%). It is important to highlight that the normal GIN threshold in adults is 5 ms or lower. The NEG performed significantly worse in this sample as it had fewer individuals in the normal range (43.3%). This indicates that the NEG had worse temporal resolution—and therefore worse auditory temporal processing—than the CG. Moreover, a 3.0 OR of having abnormal GIN test results was found by comparing individuals exposed to occupational noise with those not exposed, which suggests that this test, when used to investigate noise-induced hidden hearing loss, effectively identifies individuals at a higher risk of changes in auditory temporal processing.

The studies by Kumar et al. [25] and Liberman et al. [27] used modulation detection and duration pattern tests and verified worse results in the auditory temporal processing tests in the noise-exposed individuals than in the non-exposed ones [25,27]. It has been suggested that these results show a tendency toward decreased temporal processing abilities in noise-exposed individuals, indicating that the noise can significantly distort the processing of supraliminal temporal cues, which would make hearing even more difficult in adverse listening situations [25].

Cochlear functioning was assessed with TEOAE, according to which the NEG had lower total amplitude values and more individuals without responses. The inhibitory effect of the RE efferent pathway was assessed in the individuals with TEOAE responses. The NEG had a greater absence of this inhibitory effect and a lower percentage of inhibition. Moreover, an 8.1 OR of absent inhibitory effects of the efferent pathway was found by comparing the individuals exposed to occupational noise with those not exposed to occupational noise. This suggests that this procedure, when used to investigate noise-induced hidden hearing loss, effectively identifies those individuals at a higher risk of changes in the efferent auditory pathway.

The studies assessing normal hearing individuals exposed to noise and those not exposed to noise also found a greater absence of otoacoustic emissions in the exposed group [30,31]. This indicates that the cochlear function in noise-exposed individuals already shows signs of damage, even with normal hearing thresholds [32,33,34].

Some researchers highlight the importance of studying not only otoacoustic emission amplitudes but also the olivocochlear efferent auditory pathway, which was believed to be responsible for protecting hair cells from noise exposure. However, since the vulnerability of the synapses between hair cells and the auditory nerve was discovered, some studies have highlighted the protective role of the efferent system against synaptopathy [35,36].

In studies that assessed the inhibitory effect of the efferent auditory pathway with TEOAE suppression, the researchers found increased synaptopathy in noise-exposed guinea pigs [35] and old guinea pigs [36] after the efferent bundle section. Based on our findings and these previous studies, we hypothesize that this portion of the efferent auditory system is impaired in noise-exposed individuals, contributing to noise-induced hidden hearing loss symptoms. Therefore, this procedure should be included in the battery to assess noise-exposed individuals.

Concerning the electrophysiological assessments performed in this study, the NEG had increased latencies in wave V and interpeak intervals III-V and I-V.

A recent study conducted in 56 normal hearing adults divided into groups at low and high risk of hidden hearing loss and matched according to age found similarly increased interpeak results. The authors found increased values in interpeak intervals I–III and I–V in the group at high risk, suggesting delayed central neural conduction, which may be a consequence of hearing impairments more distally present in the auditory system. The authors also found different V/I amplitude ratios from those in the present study as the high-risk group had a higher mean value. These findings were explained as being due to a central gain control mechanism, which compensated for the reduced input. On the other hand, the authors question the clinical usefulness of the V/I amplitude ratio as it has great variability [37].

The study by Nam et al. [38] compared normal hearing individuals exposed to an episode of intense high-pitched noise with those not exposed and did not find differences in ABR latencies, amplitudes, or interpeak intervals between the groups. The authors questioned the fact that the data were collected retrospectively since it is possible that the noise-induced lesions in some individuals would not be severe enough to cause synaptopathy, which may explain the difference in findings in relation to other studies.

Concerning LLAEP, the NEG in this study had smaller P3 amplitudes in both ears. Previous research on the P3 component using verbal and nonverbal stimuli in workers exposed to occupational noise and those not exposed verified that the exposed ones had greater P3 latency values with both stimuli than those not exposed to occupational noise [39].

Brattico et al. [40] assessed workers exposed to occupational noise and compared them with a control group. They observed a smaller P1 amplitude in the noise-exposed group. The authors suggested that prolonged noise exposure changed the strength and hemispheric organization of speech sound discrimination and decreased the cortical processing speed.

A more recent study used magnetoencephalography to record cortical responses to speech stimuli with a multi-talker background noise. It verified that normal hearing individuals with speech intelligibility difficulties in noise had less cortical speech tracking than the control group [41]. These possible changes observed more rostrally in the auditory system may result from synaptopathy, which can trigger a series of neural processing changes in the posterior stages of the auditory system. This may explain some hidden hearing loss manifestations in humans [12].

Our study hypothesized that workers exposed to noise performed worse in peripheral and central assessments. In summary, this study found abnormal results in various audiological battery tests for individuals exposed to occupational noise, even with normal hearing thresholds. Hence, we suggest using the following procedures along with conventional audiometry to assess noise-induced damage early:−Speech recognition in noise, as poor performances in difficult listening conditions are the most common signs of hidden hearing loss [25,26,27,42].−Auditory temporal processing assessment, which may indicate the status of auditory processing abilities in adverse listening conditions [25,27].−Assessment of the inhibitory effect of the efferent auditory pathway, which provides indications of the efferent system protection from synaptopathy [35,36].−ABR, assessing V/I wave amplitude ratio to help identify hidden hearing loss [37,43].−LLAEP, to assess possible changes more rostrally in the auditory system [12,39].

The strengths of this study include the sample selected with extremely strict inclusion and exclusion criteria—including study and control groups matched according to age, which made it possible to control this variable as much as possible. Moreover, the study used an extensive battery of tests to assess the whole peripheral and central auditory systems and to identify which procedures were more suitable to assess noise-exposed individuals at higher risk of developing hidden hearing loss. Our findings suggest that the current noise regulation laws may not be enough to effectively protect workers exposed to occupational noise, and we verified that the exposed workers performed worse in auditory assessments than those who had not been exposed. Thus, hidden hearing loss can be a ubiquitous and neglected health problem. It is also possible that hearing protection devices are not being used properly or are not providing adequate attenuation [44,45] since these workers reported using them during their workday.

It is important to consider the fact that some of our results were different from those obtained in previous studies—which may be due to different study populations (different levels and types of noise and duration of exposure) and the different procedures used, as well as other methodological differences. Furthermore, different results in human studies may be due to the lower susceptibility of human auditory structures to adverse noise exposure effects than that of guinea pigs. Valero et al. (2017) [11] described the fact that primates need approximately 20 dB of noise more than rodents to induce a similar degree of cochlear synaptopathy, highlighting the different susceptibility between species. Therefore, noise-induced damage in humans may be lower; moreover, the procedures available for humans may not be sensitive enough to detect this damage. Furthermore, the measurements used, especially the electrophysiological ones, show great inter-subject variability, hindering their clinical use for this purpose [12].

The limitations of the study include its cross-sectional design, which prevents inferences on the causality of the changes that were found, as well as the sample size and the relatively long exposure period. Future longitudinal studies may have larger samples and divide them into different groups according to their exposure time, allowing for additional comparisons.

## 5. Conclusions

Despite the normal hearing thresholds verified with conventional audiometry, the NEG performed worse than the CG in most of the audiological assessments conducted in this study.

Therefore, it is suggested that the auditory function of normal hearing individuals exposed to occupational noise may be impaired. This highlights the importance of including complementary assessments in the battery of tests for noise-exposed individuals and of conducting further studies to investigate hidden hearing loss in this population.

## Figures and Tables

**Table 1 brainsci-13-00968-t001:** Signal-to-noise ratio (in dB) of the speech recognition threshold in noise (SRTN) and percentage of correct answers in the speech in noise test (SN) in an acoustic field, per group.

		Group	Mean	Standard Deviation	*p*-Value (ANOVA)
SNR (in dB)	RE SRTN	CG	−3.07	2.22	0.541
		NEG	−2.71	2.30	
	LE SRTN	CG	−3.00	1.66	0.653
		NEG	−3.21	1.96	
% of correct answers	SN with 0 dB SNR	CG	78.2	7.61	0.023 *
		NEG	69.8	13.67	
	SN with −10 dB SNR	CG	53.7	16.51	0.102
		NEG	43.3	19.00	

Legend: SNR—signal-to-noise ratio; RE—right ear; LE—left ear; *—significant difference.

**Table 2 brainsci-13-00968-t002:** Gap detection thresholds (in ms) and percentage of correct answers (in %) in the GIN test.

	Group	Mean	Standard Deviation	*p*-Value
GIN threshold	CG	5.20	1.24	0.116 a
	NEG	5.86	1.92	
% of correct answers	CG	69.4	9.06	0.042 a*
	NEG	64.1	10.6	
% of abnormal results	CG	20	-	0.052 b
	NEG	43.3	-	

Legend: GIN—gaps-in-noise; ms—milliseconds; a—ANOVA; b—chi-square; *—significant difference.

**Table 3 brainsci-13-00968-t003:** Amplitude in transient evoked otoacoustic emissions (TEOAE) (in dB) and percentage of ears without responses per group.

		Group	Mean	Standard Deviation	*p*-Value
Total amplitude in TEOAE	RE	CG	12.28	3.04	0.048 a*
		NEG	10.38	4.18	
	LE	CG	10.05	4.24	0.045 a*
		NEG	7.66	4.80	
% of absent responses	RE	CG	6.66	-	0.019 b*
		NEG	30	-	
	LE	CG	13.33	-	0.319
		NEG	23.33	-	

Legend: TEOAE—transient evoked otoacoustic emissions; RE—right ear; LE—left ear; a—ANOVA; b—chi-square; *—significant difference.

**Table 4 brainsci-13-00968-t004:** Inhibitory effect of the efferent auditory pathway (in micropascals) in the right ear and the percentage of individuals without the inhibitory effect per group.

	Group	Mean	Standard Deviation	*p*-Value
Inhibitory effect (µPa)	CG	3.24	0.69	0.009a*
	NEG	2.90	0.08	
% of individuals without the effect	CG (n = 29)	6.89	-	0.015b*
	NEG (n = 24)	37.5	-	

Legend: µPa—micropascal; a—ANOVA; b—Fisher’s exact; *—significant difference.

**Table 5 brainsci-13-00968-t005:** Latencies of waves I, III, and V and interpeak intervals I-III, III-V, and I-V (in ms); ABR V/I ratio per group.

		Group	Mean	Standard Deviation	*p*-Value (ANOVA)
Waves (in ms)	I—RE	CG	1.62	0.08	0.554
		NEG	1.63	0.13	
	I—LE	CG	1.62	0.10	0.458
		NEG	1.64	0.13	
	III—RE	CG	3.81	0.21	0.758
		NEG	3.82	0.15	
	III—LE	CG	3.81	0.23	0.436
		NEG	3.85	0.17	
	V—RE	CG	5.74	0.15	0.312
		NEG	5.79	0.21	
	V—LE	CG	5.74	0.17	0.017 *
		NEG	5.86	0.20	
Interpeak intervals (in ms)	I-III RE	CG	2.22	0.12	0.214
		NEG	2.17	0.14	
	I-III LE	CG	2.21	0.12	0.768
		NEG	2.20	0.16	
	III-V RE	CG	1.90	0.12	0.229
		NEG	1.95	0.18	
	III-V LE	CG	1.90	0.15	0.005*
		NEG	2.02	0.16	
	I-V RE	CG	4.12	0.15	0.441
		NEG	4.15	0.20	
	I-V LE	CG	4.12	0.18	0.040 *
		NEG	4.24	0.25	
Amplitude ratio	V/I RE	CG	1.87	1.71	0.537
		NEG	2.11	1.30	
	V/I LE	CG	2.33	2.69	0.224
		NEG	1.67	1.18	

Legend: ABR—auditory brainstem response; RE—right ear; LE—left ear; ms—milliseconds; I—wave I; III—wave III; V—wave V; I-III—interpeak interval I-III; III-V—interpeak interval III-V; I-V—interpeak interval I-V; V/I—amplitude ratio between waves V and I; *—significant difference.

**Table 6 brainsci-13-00968-t006:** LLAEP latencies (in ms) and amplitudes (µV) per group.

		Group	Mean	Standard Deviation	*p*-Value (ANOVA)
Latency (in ms)	P1 RE	CG	55.4	16.9	0.366
		NEG	52.0	11.5	
	P1 LE	CG	52.2	13.5	0.621
		NEG	50.6	11.4	
	N1 RE	CG	97.3	18.9	0.352
		NEG	93.5	12.0	
	N1 LE	CG	94.5	17.4	0.642
		NEG	92.5	14.5	
	P2 RE	CG	187.0	31.9	0.653
		NEG	183.6	27.3	
	P2 LE	CG	180.6	29.4	0.971
		NEG	180.9	27.3	
	N2 RE	CG	259.3	37.9	0.347
		NEG	268.1	34.5	
	N2 LE	CG	254.7	40.9	0.197
		NEG	269.1	45.0	
	P3 RE	CG	306.6	29.9	0.572
		NEG	311.2	32.2	
	P3 LE	CG	307.9	37.2	0.688
		NEG	312.2	45.6	
Amplitude (µV)	P1-N1 RE	CG	3.63	1.92	0.533
		NEG	3.36	1.39	
	P1-N1 LE	CG	3.84	1.93	0.373
		NEG	3.43	1.59	
	P2-N2 RE	CG	4.28	2.64	0.275
		NEG	4.98	2.30	
	P2-N2 LE	CG	4.78	3.77	1.000
		NEG	4.78	1.77	
	P3 amplitude RE	CG	9.92	4.40	0.001 *
		NEG	5.98	4.72	
	P3 amplitude LE	CG	9.10	4.57	0.002 *
		NEG	5.58	3.61	

Legend: LLAEP—long-latency auditory evoked potentials; RE—right ear; LE—left ear; ms—milliseconds; µV—microvolts; P1—positive peak 1; N1—negative peak 1; P2—positive peak 2; N2—negative peak 2; P3—positive peak 3; P1-N1—amplitude of wave P1-N1; P2-N2—amplitude of wave P2-N2; P3—amplitude of wave P3; *—significant difference.

## Data Availability

Data are unavailable due to the privacy of participants.

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
