# Peer review of "Evaluation of Subtle Auditory Impairments with Multiple Audiological Assessments in Normal Hearing Workers Exposed to Occupational Noise"

_brainsci, 2023, doi:10.3390/brainsci13060968_

Round 1

Reviewer 1 Report

This is an interesting paper and multiple methods were used to confirm subtle physiological changes in age-marched noise exposure group vs. non noise exposure group, which provide physiological evidence likely supporting author’s hypothesis -there is an association between hidden hearing loss and occupational noise exposure (even using the hearing protection in noise environment). Of note, it is still controversial if a phenomenon that has been labeled hidden hearing loss is related with synaptopathy in humans.  There are some concerns as follows: 1. Since this study has no immunohistochemistry results, the focus of this paper should not be on synaptopathy pathology and the word “synaptopathy” in title should be deleted. 2. The time of noise exposure has a large SD (6.1 years), if the authors divide subjects into different groups according to their exposure time, for example, 5, 7, 9, 11, 13 …years (occupational noise exposure), the results may be different or more striking. 3. I am not sure if this study used a very stringent approach to do the noise-exposure questionnaires since it is not shown in this paper.  4. The introduction put too much energy on description of HHL pathology, however, topics based on the major hypothesis of this paper and the fact that people with subtle hearing issue under protection in noise exposure are not easily be detected are neglected.    5. Discussion part needs to be rephrased, it is poorly organized. What I suggest is to focus on (1). Summarize your findings and provide the essential interpretations based on key findings (2). Compare and contrast to previous studies and point out your study’s limitations or any unexpected findings (concisely), such as sample size is small and variation of exposure time span  is relative large…etc. 3) Summarize your hypothesis and purpose of study, the significance, such as: regular protection in occupational noise exposure cannot effectively prevent auditory impairment since the worse performance in auditory assessments was found in noise exposure vs. non noise exposure , hidden hearing loss may be a ubiquitous health issue (neglected), hearing protection device may not be an efficient  way to prevent HHL  based on your data -Lavg 88dBA with 69% daily dose, which is less than  NIOSH standard ( 88 dBA exposure time to reach 100% noise dose would be 4 hours)…etc. Avoid unnecessary details that may distract people from a major part of your study. (4). Potential future research and clinical Implications.

This paper is not quite well written, and the associate editor may need to put some efforts to make this paper readable. 

Title: I suggest as follow in lieu of “Audiological assessment findings in normal hearing workers exposed to occupational noise: Indicators of noise-induced syn- aptopathy?”

Subtle auditory impairments scrutinized by multiple auditory assessments were found in workers exposed to occupational noise 

Line 32-34 “Noise-induced hearing loss results from lesions or dysfunctions, especially in cochlear outer hair cells, which were believed until recently to be the structures most vulnerable to noise”, “until recently” need to be deleted.

Line 41-42 “Furthermore, this synaptopathy is specific to the high-thresholds fibers that form synapses on the modiolar side of the inner hair cells, whose low spontaneous rates”. I guess it should be “Furthermore, this synaptopathy is specific located in the high-thresholds fibers that form synapses on the modiolar side of the inner hair cells, whose spontaneous rates are relatively low”.

Line 42-46 “They are particularly important in noisy environments, where fibers with high trigger rates (on the pillar side of the inner hair cells) saturate more easily. Hence, it is believed that hearing thresholds are not impaired in quiet, whereas in noise speech intelligibility and auditory temporal processing would be difficult”.

Don’t know what the authors want to express other than English issues. I recommend reading the paper by Furman et al. (J Neurophysiol 2013), then, rephrase this part. Guess that the authors want to point out a theory that some people believe: neurons with the highest thresholds may generate a complementary information pool by receiving predominantly low-SR auditory nerve fiber output and adapting their thresholds higher under loud conditions- a base of difficulty in speech intelligibility and temporal processing.

Line 49-52 “Since this cascade of events could determine alterations throughout the entire auditory system, it would be important to include procedures to assess the peripheral and central auditory pathways, including procedures that assess the most rostral portion of the pathway, such as long-latency evoked potentials”. It would be more adequate as “Since this cascade of events could determine alterations throughout the entire auditory system, alongside utilizing peripheral auditory assessments, it would be important to include procedures to assess the central auditory pathways such as long-latency evoked potentials for assessment of the most rostral portion of the pathway”.

Line 56-60 “Despite the large number of non-invasive measures that could potentially identify synaptopathy in humans, there is still no consensus on which neurophysiological evidence observed in animal models would also be evident in humans, which would be the underlying cause of difficulties in speech intelligibility in noise in subjects with normal audiograms”. It would be sound smooth and resonable like “Despite the large effort on experiments in specific mice strains to identify candidate assays  of non-invasive measures that could potentially identify synaptopathy in humans, there is still no consensus on which neurophysiological evidence observed in animal models would also be evident in humans, which underlies the cause of difficulties in speech intelligibility in subjects with normal audiograms after occupational noise exposure”.

Line 61-65 “Therefore, it is highly important to investigate and include other forms of assessment to monitor noise-exposed workers because conventional pure-tone audiometry is insufficient to assess their auditory system. Moreover, this type of impairment must be confirmed in humans, due to its possible practical consequences, including occupational regulations regarding noise exposure limits”.

Don’t know what the authors want to express, I guess the authors want to emphasis it is important to use multiple assessments for the investigation in noise-exposed workers since conventional audiometry test may present normality, which may cover the fact that the noise-exposed workers have subtle auditory issue, such as struggling to follow a conversation in noisy environments?

Line 66-70 “The hypothesis of our study is that workers exposed to noise will perform worse in peripheral and central assessments. As a differential of the present study, we highlight the pairing by age of the groups exposed and not exposed to noise and the use of a large number of audiological procedures that allow evaluating different portions of the peripheral and central auditory pathway”.

My suggestions:” The hypothesis of our study is that workers exposed to noise even under the hearing protection may have unnoticed auditory impairment, which can detected by multiple peripheral and central auditory assessments. We paired the groups by age match with exposed or not exposed to noise, and used multiple audiological procedures that allow us to evaluate different portions of the peripheral and central auditory pathways”.

Line 71 -72 delete.

Line 80 It would be clearer that “Study Group (SG)” is changed to “Noise Exposure Group (NEG)”, thereafter.

Materials and Methods

Line 97-98 “All of them use hearing protection during their workday”, you need to do more detail about hearing protection device attenuation, such as what kind of devices they used, earplug or earmuff? and if these devices meet the standards of occupational noise exposure (if have), how often they wore…etc. I recommend reading the paper written by Grinn et al. (Frontiers in Neuroscience 2022), it is very important since your results suggest that conventional hearing protection is not able to prevent subtle hearing impairment, which is quite in contrast to common sense of public health about auditory prevention.

Line 95 “Lavg”- The average sound pressure level over a period of time (Lavg).

LLAEP is a very important method used by this study. Considering the extensive clinical applications and the significant intra / interindividual variation, the way to describe the procedure is a little simple. What is the orientation of electrodes (mastoid)? Impedance of electrodes? What are the Non-verbal stimuli like (may be like ga/,/da/e/di… I guess). Did you do normalization of protocol ? Latency reference values? Performed the analysis by waveform subtraction?

Discussion

Line 344-354  “ new procedures should be considered to assess noise-induced synaptopathy …”

The authors just listed the experiments they did in this paper and neglected the fact that they described in introduction-“there is still no consensus on which neurophysiological evidence observed in animal models would also be evident in humans”. Therefore, I recommend reading the papers written by Ridley et al (Ear Hear, 2018:39: 829-844), Valderrama et al (Front Neurosci, 2022 Sep 15;16:1000304), Grinn et al (Front Neurosci. 2022 Oct 26;16:1005148) and Buran et al (J Acoust Soc Am. 2022 Jan;151(1):561). These papers provide good hints for future directions of studies in HHL.

Author Response

COVER LETTER

We would like to thank you for the opportunity to resubmit the revised version of our manuscript entitled “Subtle auditory impairments evaluated with multiple audiological assessments in normal hearing workers exposed to occupational noise” for consideration for publication in the Brain Science. We are thankful to the referees and the Editor for their thoughtful suggestions that helped to strengthen our manuscript. This new version of the manuscript included changes in the manuscript following reviewers’ recommendations. We have addressed all the reviewers’ concerns and provided a detailed response below. To facilitate the process, we have transcribed all questions raised and answered them point by point. In addition, we are attaching a marked version of the manuscript.

Please do not hesitate to contact us if you have any further questions. We look forward to hearing back from you soon.

Reviewer 1

Comments and Suggestions for Authors

This is an interesting paper and multiple methods were used to confirm subtle physiological changes in age-marched noise exposure group vs. non noise exposure group, which provide physiological evidence likely supporting author’s hypothesis -there is an association between hidden hearing loss and occupational noise exposure (even using the hearing protection in noise environment). Of note, it is still controversial if a phenomenon that has been labeled hidden hearing loss is related with synaptopathy in humans. 

Response: Thank you. We appreciate your positive feedback.

There are some concerns as follows:

  1. Since this study has no immunohistochemistry results, the focus of this paper should not be on synaptopathy pathology and the word “synaptopathy” in title should be deleted.

Response: Thank you. We agree with the reviewer and have excluded synaptopathy in the title and most of the manuscript, changing it to hidden hearing loss.

  1. The time of noise exposure has a large SD (6.1 years), if the authors divide subjects into different groups according to their exposure time, for example, 5, 7, 9, 11, 13 …years (occupational noise exposure), the results may be different or more striking.

Response: As the group exposed to noise was composed of 30 individuals, making this subdivision would make the subgroups very small, impairing the analyses. We included this limitation in discussion.

  1. I am not sure if this study used a very stringent approach to do the noise-exposure questionnaires since it is not shown in this paper.  

Response:

- In relation to the current occupational exposure, in addition to the questionnaire made with the worker, we have the information by Environmental Risks Prevention Program that includes and quantifies the risks that workers are exposed to.

- Previous risk – questions about occupational history with exposure to noise (Function, Field of activity, Duration, HPD use and type)

- Non-occupational history with exposure to noise (noisy hobby, use of motorcycle, use of firearm, concerts/parties/stadiums/racings, explosion, etc.). If yes, description of items.

  1. The introduction put too much energy on description of HHL pathology, however, topics based on the major hypothesis of this paper and the fact that people with subtle hearing issue under protection in noise exposure are not easily be detected are neglected.   

Response: Thank you. We included more information about this topic and another references.

  1. Discussion part needs to be rephrased, it is poorly organized. What I suggest is to focus on (1). Summarize your findings and provide the essential interpretations based on key findings (2). Compare and contrast to previous studies and point out your study’s limitations or any unexpected findings (concisely), such as sample size is small and variation of exposure time span  is relative large…etc. 3) Summarize your hypothesis and purpose of study, the significance, such as: regular protection in occupational noise exposure cannot effectively prevent auditory impairment since the worse performance in auditory assessments was found in noise exposure vs. non noise exposure , hidden hearing loss may be a ubiquitous health issue (neglected), hearing protection device may not be an efficient  way to prevent HHL  based on your data -Lavg 88dBA with 69% daily dose, which is less than  NIOSH standard ( 88 dBA exposure time to reach 100% noise dose would be 4 hours)…etc. Avoid unnecessary details that may distract people from a major part of your study. (4). Potential future research and clinical Implications.

Response: Thank you. We have reorganized the discussion in an attempt to make it clearer. 

This paper is not quite well written, and the associate editor may need to put some efforts to make this paper readable. 

Response: We carry out an English review to improve the text.

Title: I suggest as follow in lieu of “Audiological assessment findings in normal hearing workers exposed to occupational noise: Indicators of noise-induced syn- aptopathy?”

 Subtle auditory impairments scrutinized by multiple auditory assessments were found in workers exposed to occupational noise 

Response: The two reviewers proposed changes to the title. Therefore, we made a modification seeking to include both suggestions.

Line 32-34 “Noise-induced hearing loss results from lesions or dysfunctions, especially in cochlear outer hair cells, which were believed until recently to be the structures most vulnerable to noise”, “until recently” need to be deleted.

Response: The suggestion was accepted.

Line 41-42 “Furthermore, this synaptopathy is specific to the high-thresholds fibers that form synapses on the modiolar side of the inner hair cells, whose low spontaneous rates”. I guess it should be “Furthermore, this synaptopathy is specific located in the high-thresholds fibers that form synapses on the modiolar side of the inner hair cells, whose spontaneous rates are relatively low”.

Response: The suggestion was accepted.

Line 42-46 “They are particularly important in noisy environments, where fibers with high trigger rates (on the pillar side of the inner hair cells) saturate more easily. Hence, it is believed that hearing thresholds are not impaired in quiet, whereas in noise speech intelligibility and auditory temporal processing would be difficult”.

Don’t know what the authors want to express other than English issues. I recommend reading the paper by Furman et al. (J Neurophysiol 2013), then, rephrase this part. Guess that the authors want to point out a theory that some people believe: neurons with the highest thresholds may generate a complementary information pool by receiving predominantly low-SR auditory nerve fiber output and adapting their thresholds higher under loud conditions- a base of difficulty in speech intelligibility and temporal processing.

Response: Thank you. We rewrite the paragraph in an attempt to make it clearer. 

Line 49-52 “Since this cascade of events could determine alterations throughout the entire auditory system, it would be important to include procedures to assess the peripheral and central auditory pathways, including procedures that assess the most rostral portion of the pathway, such as long-latency evoked potentials”. It would be more adequate as

“Since this cascade of events could determine alterations throughout the entire auditory system, alongside utilizing peripheral auditory assessments, it would be important to include procedures to assess the central auditory pathways such as long-latency evoked potentials for assessment of the most rostral portion of the pathway”.

Response: The suggestion was accepted.

Line 56-60 “Despite the large number of non-invasive measures that could potentially identify synaptopathy in humans, there is still no consensus on which neurophysiological evidence observed in animal models would also be evident in humans, which would be the underlying cause of difficulties in speech intelligibility in noise in subjects with normal audiograms”. It would be sound smooth and resonable like “Despite the large effort on experiments in specific mice strains to identify candidate assays  of non-invasive measures that could potentially identify synaptopathy in humans, there is still no consensus on which neurophysiological evidence observed in animal models would also be evident in humans, which underlies the cause of difficulties in speech intelligibility in subjects with normal audiograms after occupational noise exposure”.

Response: The suggestion was accepted.

Line 61-65 “Therefore, it is highly important to investigate and include other forms of assessment to monitor noise-exposed workers because conventional pure-tone audiometry is insufficient to assess their auditory system. Moreover, this type of impairment must be confirmed in humans, due to its possible practical consequences, including occupational regulations regarding noise exposure limits”.

Don’t know what the authors want to express, I guess the authors want to emphasis it is important to use multiple assessments for the investigation in noise-exposed workers since conventional audiometry test may present normality, which may cover the fact that the noise-exposed workers have subtle auditory issue, such as struggling to follow a conversation in noisy environments?

Response: Thank you. We rewrite the paragraph in an attempt to make it clearer. 

Line 66-70 “The hypothesis of our study is that workers exposed to noise will perform worse in peripheral and central assessments. As a differential of the present study, we highlight the pairing by age of the groups exposed and not exposed to noise and the use of a large number of audiological procedures that allow evaluating different portions of the peripheral and central auditory pathway”.

My suggestions:” The hypothesis of our study is that workers exposed to noise even under the hearing protection may have unnoticed auditory impairment, which can detected by multiple peripheral and central auditory assessments. We paired the groups by age match with exposed or not exposed to noise, and used multiple audiological procedures that allow us to evaluate different portions of the peripheral and central auditory pathways”.

Response: The suggestion was accepted.

Line 71 -72 delete.

Response: The suggestion was accepted.

Line 80 It would be clearer that “Study Group (SG)” is changed to “Noise Exposure Group (NEG)”, thereafter.

Response: The suggestion was accepted.

Materials and Methods

Line 97-98 “All of them use hearing protection during their workday”, you need to do more detail about hearing protection device attenuation, such as what kind of devices they used, earplug or earmuff? and if these devices meet the standards of occupational noise exposure (if have), how often they wore…etc. I recommend reading the paper written by Grinn et al. (Frontiers in Neuroscience 2022), it is very important since your results suggest that conventional hearing protection is not able to prevent subtle hearing impairment, which is quite in contrast to common sense of public health about auditory prevention.

Response: Workers use silicone earplug (90%) or muffin (10%), as instructed by the occupational health team at the university where the study was carried out. Workers refer to using it during their working hours. However, as mentioned in the discussion, even using HPD with attenuations that provide “theoretical” safe noise levels, this does not guarantee that this attenuation will be achieved in the individual's ear (Takada et al, 2020; Morata et al, 2021). Unfortunately, fit testing is not a worldwide mandated practice required by law, with the exception of Canada. In this way, it is not possible to guarantee the attenuation described by the manufacturer and thus, even with the use of HPD, the protection of the worker may not be effective.

Line 95 “Lavg”- The average sound pressure level over a period of time (Lavg).

Response: The suggestion was accepted.

LLAEP is a very important method used by this study. Considering the extensive clinical applications and the significant intra / interindividual variation, the way to describe the procedure is a little simple. What is the orientation of electrodes (mastoid)? Impedance of electrodes? What are the Non-verbal stimuli like (may be like ga/,/da/e/di… I guess). Did you do normalization of protocol ? Latency reference values? Performed the analysis by waveform subtraction?

Response: The suggestion was accepted. The information was included in Method section, with the exception of the latency reference values, since they were not used, since the results were not classified as normal or altered.

Discussion

Line 344-354  “ new procedures should be considered to assess noise-induced synaptopathy …”

The authors just listed the experiments they did in this paper and neglected the fact that they described in introduction-“there is still no consensus on which neurophysiological evidence observed in animal models would also be evident in humans”. Therefore, I recommend reading the papers written by Ridley et al (Ear Hear, 2018:39: 829-844), Valderrama et al (Front Neurosci, 2022 Sep 15;16:1000304), Grinn et al (Front Neurosci. 2022 Oct 26;16:1005148) and Buran et al (J Acoust Soc Am. 2022 Jan;151(1):561). These papers provide good hints for future directions of studies in HHL.

Response: Yes, we agree with reviewer, there is no consensus, but based on our findings and on some other previous studies, we suggest that these procedures be considered as indicators of signs of alteration in the auditory system resulting from damage caused by noise, even in the presence of normal hearing. We modified the writing a little and added other references to make it clearer.

Reviewer 2 Report

The present study aims to investigate the auditory system of normal hearing workers exposed to occupational noise in attempts to disclose synaptopathy. 

60 participants were recruited, 30 for the study group (SG) and 30 for the control group (CG). The study groups contained individuals between 18-50 years of age and all with normal hearing thresholds. The SG was composed of individuals working with maintenance and exposed to occupational noise the last 5 years and the CG consisted of individuals not exposed to occupational noise.

The participants were exposed to a large battery of audiological tests.

Sentence recognition threshold in noise test and speech recognition index in silence test did not give any significant difference between the groups, whereas speech recognition test in noise differed significantly. The gaps-in-noise test resulted in higher detection values in the SG compared to CG. When it comes to transient evoked otoacoustic emissions of SG had  lower amplitudes than those of CG. When performing ABR the wave V latencies were prolonged as well as the interpeak intervals of III-V and I-V. Regarding long-latency auditory evoked potentialsthey did not show any difference regarding latencies but the amplitudes of P3 were lower in the SG compared to CG.

It is concluded that SG performed worse than CG in most audiological assessments conducted in this study. Thus, it is suggested that the auditory function of normal hearing individuals exposed to occupational noise is impaired. This highlights the importance of including complementary assessments in the battery of tests for noise-exposed individuals and further studies to investigate synaptopathy in this population. 

In summary this study is interesting though it does not give any substantial support to the concept synaptopathy. The study is well designed and well performed under controlled conditions. It pays attention to the need of investigation in also individuals with normal hearing thresholds exposed to well defined “portions” of noise.  I am ready to recommend this paper for publication in Brain Sciences but prior to that the questions/comment below should be answered.

·      The population exposed to noise is well defined as workers exposed to occupational noise obtained in their position as maintenance workers at the university. In contrast the CG is not well described which kind of work they participated in and if they also were working at the university.

·      Our current definition of synaptopathy refers to cochlear synaptopathy. With the audiological tests used in the present study do we really have any clearcut evidence that the differences seen is based on synaptopathy?

·      The title of this manuscript reads “Audiological assessment findings in normal hearing workers exposed to occupational noise: Indicators of noise-induced synaptopathy? “ Do the tests used really indicate synaptopathy? And would it be more adequate to use a title reading “Audiological assessment findings in normal hearing workers exposed to occupational noise” ?

Author Response

The present study aims to investigate the auditory system of normal hearing workers exposed to occupational noise in attempts to disclose synaptopathy.

60 participants were recruited, 30 for the study group (SG) and 30 for the control group (CG). The study groups contained individuals between 18-50 years of age and all with normal hearing thresholds. The SG was composed of individuals working with maintenance and exposed to occupational noise the last 5 years and the CG consisted of individuals not exposed to occupational noise.

The participants were exposed to a large battery of audiological tests.

Sentence recognition threshold in noise test and speech recognition index in silence test did not give any significant difference between the groups, whereas speech recognition test in noise differed significantly. The gaps-in-noise test resulted in higher detection values in the SG compared to CG. When it comes to transient evoked otoacoustic emissions of SG had  lower amplitudes than those of CG. When performing ABR the wave V latencies were prolonged as well as the interpeak intervals of III-V and I-V. Regarding long-latency auditory evoked potentials, they did not show any difference regarding latencies but the amplitudes of P3 were lower in the SG compared to CG.

It is concluded that SG performed worse than CG in most audiological assessments conducted in this study. Thus, it is suggested that the auditory function of normal hearing individuals exposed to occupational noise is impaired. This highlights the importance of including complementary assessments in the battery of tests for noise-exposed individuals and further studies to investigate synaptopathy in this population.

In summary this study is interesting though it does not give any substantial support to the concept synaptopathy.

Response: Thank you. We appreciate your positive feedback. Thank you for the opportunity to submit the revised version of our manuscript. We agree and changed the term to hidden hearing loss.

The study is well designed and well performed under controlled conditions. It pays attention to the need of investigation in also individuals with normal hearing thresholds exposed to well defined “portions” of noise.  I am ready to recommend this paper for publication in Brain Sciences but prior to that the questions/comment below should be answered.

  • The population exposed to noise is well defined as workers exposed to occupational noise obtained in their position as maintenance workers at the university. In contrast the CG is not well described which kind of work they participated in and if they also were working at the university.

Response: Thank you. We included this information in Methods section.

  • Our current definition of synaptopathy refers to cochlear synaptopathy. With the audiological tests used in the present study do we really have any clearcut evidence that the differences seen is based on synaptopathy?

Response: No, actually the reviewer is right, there is no clear evidence, only hypotheses based on animal studies, since there is no immunohistochemistry evidence. Therefore, we chose to change the terminology throughout the article to hidden hearing loss, also according to reviewer 1's suggestion.

  • The title of this manuscript reads “Audiological assessment findings in normal hearing workers exposed to occupational noise: Indicators of noise-induced synaptopathy? “ Do the tests used really indicate synaptopathy? And would it be more adequate to use a title reading “Audiological assessment findings in normal hearing workers exposed to occupational noise” ?

Response: The two reviewers proposed changes to the title. Therefore, we made a modification seeking to include both suggestions.

Round 2

Reviewer 1 Report

The first revision still has  issues, even though some of issues were gone according to the reviewer’s suggestions.

Title:

I recommend as follows: “Evaluation of subtle auditory impairments by multiple audiological assessments on normal hearing workers exposed to occupational noise”.

Line 13-14: “The aim was to investigate the auditory system of normal hearing workers exposed to occupational noise”. It would be more accurate as “The aim was to investigate the auditory function of  workers exposed to occupational noise with normal hearing by multiple audiological assessments”.

Line 21-22:” NEG performed worse than CG in SNR 0 (p-value 0.023). In GIN, NEG 21 had a significantly lower percentage of correct answers (p-value 0.042)”.

First “SNR” appeared in this paper, it should be like” signal-to-noise ratio (SNR)”.

Line 52-55: “Since this cascade of events can cause changes throughout the entire auditory system, not only peripheral but also central auditory pathway assessments (such as long-latency evoked potentials (LLAEP)) should be used to verify the most rostral portion of the pathway”. It would be more adequate like” Since this cascade of events can cause changes throughout the entire auditory system including peripheral and central auditory pathways, special assessment, such as long-latency evoked potentials (LLAEP), should be used to verify the most rostral portion of the auditory system”.

The discussion part needs to be revised carefully, especially pay attention to the grammar and please do the sentence by sentence checking. Most of them have issues (grammar and words organization).

Author Response

We would like to thank you for the opportunity to resubmit the revised version of our manuscript entitled “Evaluation of subtle auditory impairments with multiple audiological assessments in normal hearing workers exposed to occupational noise” for consideration for publication in the Brain Science. We are thankful to the referees and the Editor for their thoughtful suggestions that helped to strengthen our manuscript. This new version of the manuscript included changes in the manuscript following reviewers’ recommendations. We have addressed all the reviewers’ concerns and provided a detailed response below. To facilitate the process, we have transcribed all questions raised and answered them point by point. In addition, we are attaching a marked version of the manuscript.

Please do not hesitate to contact us if you have any further questions. We look forward to hearing back from you soon.

Reviewer 2

The first revision still has  issues, even though some of issues were gone according to the reviewer’s suggestions.

Response: Thank you for the opportunity to submit the revised version of our manuscript.

Title:

I recommend as follows: “Evaluation of subtle auditory impairments by multiple audiological assessments on normal hearing workers exposed to occupational noise”.

Response: Thank you. The suggestion was accepted.

Line 13-14: “The aim was to investigate the auditory system of normal hearing workers exposed to occupational noise”. It would be more accurate as “The aim was to investigate the auditory function of workers exposed to occupational noise with normal hearing by multiple audiological assessments”.

Response: Thank you. The suggestion was accepted.

Line 21-22:” NEG performed worse than CG in SNR 0 (p-value 0.023). In GIN, NEG 21 had a significantly lower percentage of correct answers (p-value 0.042)”.

First “SNR” appeared in this paper, it should be like” signal-to-noise ratio (SNR)”.

Response: Thank you. We included the information.

Line 52-55: “Since this cascade of events can cause changes throughout the entire auditory system, not only peripheral but also central auditory pathway assessments (such as long-latency evoked potentials (LLAEP)) should be used to verify the most rostral portion of the pathway”. It would be more adequate like” Since this cascade of events can cause changes throughout the entire auditory system including peripheral and central auditory pathways, special assessment, such as long-latency evoked potentials (LLAEP), should be used to verify the most rostral portion of the auditory system”.

Response: Thank you. The suggestion was accepted.

Comments on the Quality of English Language

The discussion part needs to be revised carefully, especially pay attention to the grammar and please do the sentence by sentence checking. Most of them have issues (grammar and words organization).

Response: Thank you. The text was resubmitted for English revision and we request special attention to the discussion.
